# Firearms and violence in Europe–A systematic review

**Katharina Krüsselmann**[ID]*, **Pauline Aarten**, **Marieke Liem**

Institute of Security and Global Affairs, Leiden University, Leiden, The Netherlands

☯ These authors contributed equally to this work.
* k.krusselmann@fgga.leidenuniv.nl

## Abstract

### Background

Higher availability of firearms has been connected to higher rates of interpersonal violence in previous studies. Yet, those studies have focused mainly on the United States, or used aggregated international data to study firearm violence. Whether those aggregated findings are applicable to understanding the phenomenon in continental Europe specifically remains unclear. The aim of this systematic review is to bring together all studies that exclusively use European data.

### Methods

Nine databases were searched, resulting in more than 1900 individual studies. These studies were assessed on relevance and eligibility for this study, based on their title, abstract and full text. Information on study characteristics, operationalizations of main concepts and study results were extracted from the six eligible studies.

### Results

Four studies assessed the impact of firearm restrictive regulations on the rate of firearm homicides. Two other studies correlated rates of firearm availability and -violence. Results vary: some studies show a clear decline once availability of firearms is restricted, while others indicate a limited effect on only a very specific subgroup, such as female victims, or national guards with weapons at home. Moreover, studies used various operationalizations for firearm availability, thereby decreasing the comparability of findings.

### Conclusion

Empirical research exclusively using European data is still lacking. To increase comparability of future studies, methodological inconsistencies and regional gaps need to be overcome. Assessing how firearm availability can be measured with reliable and valid proxies across countries will be a crucial first step to improve future research on the link between firearms and firearm violence.

**Data Availability Statement:** All relevant data are within the paper and its Supporting Information files.

**Funding:** M.L. received funding from the European Commission Directorate-General for Migration and Home Affairs (Grant number 867477) for Project

TARGET, which informed parts of this study and financed K.K.'s work on this study. Moreover, the publication fee of this study has been funded by the same source. The funders had no role in study design, data collection and analysis, decision to publish, or preparation of the manuscript.

**Competing interests:** The authors have declared that no competing interests exist.

## Introduction

It is estimated that around 7000 people (0.9 per 100.000 population) die of gunshot wounds each year in continental Europe, including suicides, unintentional accidents involving firearms, and criminal homicides [1]. Although many types of weapons can cause death or bodily harm, firearms are of specific interest when studying violence, due to their high lethality, widespread use on a global scale and value for criminals [2]. How firearms are linked specifically to violent death has been studied extensively, but existing studies heavily focus on the US context, where more than 12 people per 100.000 population die of gunshot wounds each year [3, 4]. Most research to date focuses on the firearm availability hypothesis, which assumes that an increase in firearm availability leads to an increase in violent crime [5–7]. Yet, even with these existing studies, as causal links between the prevalence of firearms and violence remain unclear, heated discussions on the connection between the two phenomena continue both in academia and beyond [8].

The question arises whether findings from US-based studies are applicable to other global regions, such as Europe, given variations in existing gun cultures and firearm legislations. In contrast to the US, where the right to own firearms is implemented in the Constitution under the Second Amendment, European countries have strict regulations that mostly ban civilians from bearing guns, with only few exceptions. Member states of the European Union follow the same framework of regulations regarding civilian access to firearms, which leaves room for proportionate national variations between the member states [9, 10]. Such national variations between member states seem irreconcilable, given the difference in gun culture not just on a global scale, but also amongst European countries, as exemplified by the challenging approval of the 2017 Firearm Directive in the European Council in which Poland and the Czech Republic voted against the amendment for being too strict, whereas Luxembourg wished for harsher restrictions [10]. In addition, Europe does not only differ significantly from the US in terms of existing gun cultures and firearms legislation [11], but also in terms of other cultural and socio-economic factors, such as the overall crime rate or levels of inequality [12]. Such factors have been found to act as moderators in violent crime rates [13, 14]. As these factors vary across countries, the validity of findings from US studies for the European context could be questioned. Therefore, a review of empirical studies testing a potential link exclusively based on European data is required.

With this systematic review, we aim to inspect existing studies that empirically examine the link between firearm availability and firearm-enabled interpersonal violence in Europe, since such an assessment does not yet exist to the best of our knowledge. To increase comparability of studies and their findings, we focus particularly on criminal forms of interpersonal violence, thus excluding firearm-enabled suicides or accidental fatal and non-fatal injuries. With the findings of our study, we seek to inform researchers, practitioners and policymakers in the domains of public health and criminal justice about the current state of knowledge regarding the association between firearm availability and violence. Furthermore, we aim to identify the lacunae of knowledge and the methodological challenges which can be addressed by future research.

## Methods

This study made use of PRISMA guidelines for conducting systematic review [15].

### Eligibility criteria

For this systematic review, we included studies that empirically examined a potential link between firearm availability and firearm enabled crimes, including homicide, non-fatal

assaults or robberies. Studies that focused on the relationship between firearm enabled crimes and other factors, such as mental illness, but included a measurement for firearm availability were eligible as well. We excluded non-criminal forms of violence, such as suicide by gunshot or accidental shootings as we expected that those forms of violence could have underlying explanatory variables that are different to criminal forms of violence and should therefore be studied separately. Furthermore, we excluded studies that did not examine the link between measurements of those two main concepts in a statistical manner to eliminate potential subjectivity from our synthesis of findings. Studies that did not explicitly differentiate firearms from other types of weapons (e.g. knives) were also excluded to enhance comparability of the eligible studies, but when a clear differentiation was provided, we included the studies.

Concerning our geographical focus, we only included studies that are based on continental Europe (excluding Turkey and Russia, which are countries situated on two continents). US-based studies and studies that combined data from non-European and European origin, which inhibited us to assess data sources independently, were not further examined. When we were able to isolate the results of European data from non-European data, we included the studies.

All studies published after 1991 were selected, as 1991 marks the year the first European directive in firearm acquisition and possession was introduced in the European Union [16]. This cut-off was chosen as it can be expected that most countries on the European continent would have been following similar definitions of and regulations for firearm restrictions since then. Our expectations were that this directive might have led to empirical cross-national studies relevant for this research. Studies were excluded when they were published before 1991 or when they used data only from years before 1991. Included studies had to be written in English, German or Dutch, due to the researchers' capabilities of understanding those languages.

## Data sources

Peer-reviewed academic articles, books, book chapters, and (unpublished) doctoral dissertations were included. We searched seven databases that cover academic studies in the relevant disciplines of public health, sociology and criminology: Criminal Justice Abstracts, Embase, MedLine, PsyInfo, PubMed, Sociological Abstracts and Web of Science. In addition, we searched ProQuest and EThOS for relevant (unpublished) doctoral dissertations. If results from a dissertation were also published in the form of an academic article, the latter was chosen. Additional records were found by searching the bibliographies of relevant studies. To overcome the danger of publication bias, we emailed relevant researchers in the field and conducted a web search using Google and Google Scholar search engines on 21 April 2020 to find grey literature, such as research reports. The results of those searches were filtered based on the eligibility criteria.

## Search process

The search queries used to find relevant studies include terms concerning firearms, their prevalence, and firearm violence. Previous published systematic reviews on firearm availability in the non-European context served as an inspiration for the chosen search terms (2,6). In particular, three specific search queries were used in each of the nine databases:

◦ firearm AND availability AND violen*

◦ (firearm OR gun) AND (availability OR access) AND (violen* OR homicide)

◦ (firearm OR gun) AND (availability OR access) AND (violen* OR crim*)

## Study selection

Databases were searched between February and April 2020, using the three search queries. As shown in Fig 1, a total of 8,179 studies in the nine databases were found and reduced to 238 studies by elimination of duplicates and screening of titles and abstracts on relevance. The full texts of the remaining 238 studies were assessed on relevance using the eligibility criteria. Authors of studies missing full text were emailed with the question to provide full access to their study, resulting in ten additional studies that were included in the assessment. For 23 studies, the author did not respond, or no contact details were available or found online. In total, 232 studies were excluded during that process because they missed full-text (n = 23), discussed irrelevant topics (n = 14), did not focus exclusively on Europe (n = 107), missed empirical data (n = 45), or data measuring either firearm-enabled crimes (n = 13) or firearm availability (n = 30). As a result, six studies were included in the synthesis of results for this systematic review (Fig 1).

The quality of each eligible study was assessed based on six factors which have been used in a previous, similar systematic review about firearm laws and firearm homicides in the US [17]. Next to the study design, the authors of that review evaluated studies based on five additional measurements:

"(1) Were appropriate data source(s) and outcomemeasure(s) used for the study question?

(2) Was the time frame studied adequate (eg, sufficient surveillance before and after a law)?

(3) Were appropriate statistical tests used?

(4) Were the results robust to variations in the variables and analyses?

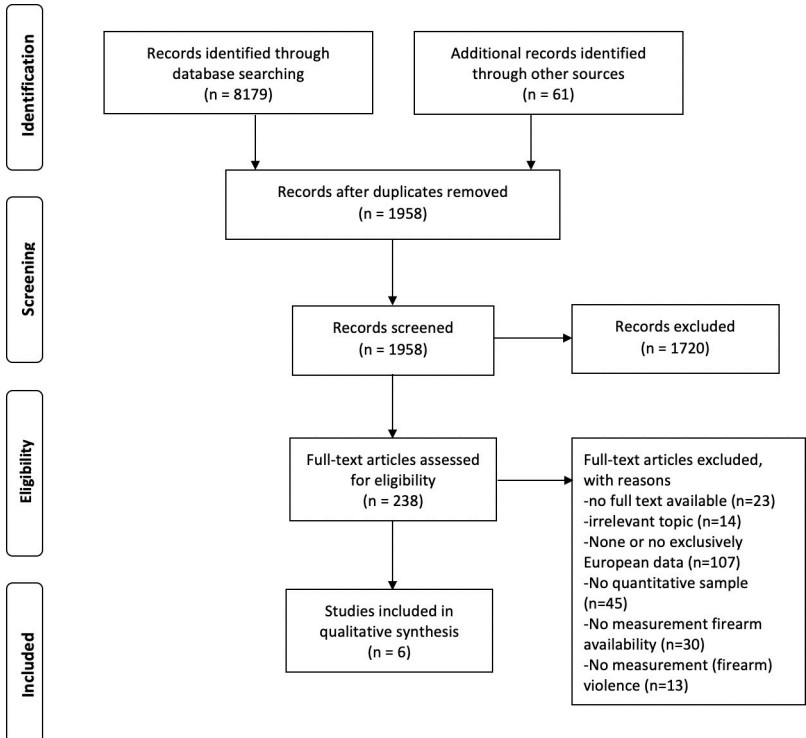

**Fig 1. Flow chart of systematic search.**

(5) Were the disaggregated data and results of control variables consistent with the literature?" [17 p108].

A study's quality was qualified as *good* when they scored high on all those elements, and as *fair* when they scored high on three to four of the factors. Studies were considered *poor* if they scored on only two or less factors. We have followed the same quality assessment.

## Results

### Study characteristics

In total, only six studies remained eligible for this study. The background information on each of those studies can be found in Table 1.

Five studies were published in peer-reviewed journals [18–22] and one report was published by researchers associated with the Flemish Peace Institute in Belgium [1]. Two studies were published between 2000 and 2010 [20, 21]. Four studies were published between 2010 and 2020 [1, 18, 19, 22]. Four of the six studies limit their research to examining the link between access to firearms and firearm-enabled violence to one country [18, 20–22]. Those studies originate from Austria (n = 2) [20, 22], Norway (n = 1) [18] and Switzerland (n = 1) [21]. The other studies include data from 16 [19] up to 33 countries [1]. Four studies use non-aggregated data [18, 20–22], whereas the two other studies rely on aggregated data from the national level, namely national homicide rates [1, 19]. All studies have scored as either fair or good in the study quality assessment.

**Table 1. Characteristics of included studies, including study locations, time frame studied, study design, sample, quality and type of study.**

| Study | Location | Time Frame | Self-reported Study Design & Sample | Quality Assessment | Type |
|---|---|---|---|---|---|
| Duquet & van Alstein (2015) 'Firearms and violent death in Europe' [1] | Austria, Belgium, Bulgaria, Croatia, Cyprus, Czech Republic, Denmark, Estonia, Finland, France, Germany, Hungary, Iceland, Ireland, Italy, Latvia, Lithuania, Luxembourg, FYR Macedonia, Malta, Moldova, Montenegro, Netherlands, Norway, Poland, Portugal, Romania, Serbia, Slovakia, Slovenia, Spain, Sweden, United Kingdom | 2007 | Cross-sectional correlational design; 33 European Nations | Fair | Report |
| Gjertsen et al. (2014) ' Mixed impact of firearms restrictions on fatal firearm injuries in males: a national observational study' [18] | Norway | 1969–2009 | Cross-sectional time series design; 434 cases of male accidental and homicidal deaths caused by firearms | Fair | Journal Article |
| Hurka & Knill (2018) 'Does regulation matter? A cross-national analysis of the impact of gun policies on homicide and suicide rates' [19] | Austria, Belgium, England and Wales, Denmark, Finland, France, Germany, Greece, Ireland, Italy, Netherlands, Norway, Portugal, Spain, Sweden, Switzerland | 1980–2010 | Cross-sectional time series design; 16 West European Nations | Good | Journal Article |
| Kapusta et al. (2007) ' Firearm legislation reform in the European Union: impact on firearm availability, firearm suicide and homicide rates in Austria' [20] | Austria | 1985–2005 | Longitudinal time series design; Firearm homicide rate 1985–2005 | Fair | Journal Article |
| Killias & Haas (2002) ' The role of weapons in violent acts: Some results of a Swiss national cohort study ' [21] | Switzerland | 1997 | Cross-sectional correlational design, 21.315 surveys by Swiss male soldiers | Fair | Journal Article |
| König et al. (2018) ' Austrian firearm legislation and its effects on suicide and homicide mortality: A natural quasi-experiment amidst the global economic crisis' [22] | Austria | 1985–2016 | Cross-sectional time series design; firearm homicide rate 1985–2016 | Fair | Journal Article |

## Outcomes

An overview of the operationalizations used for firearm availability and violence, as well as the outcomes of each of the studies can be found in Table 2. The outcomes are presented in three subsections: results from national studies on the impact of firearm legislation of -violence, cross-national studies regarding such an impact and correlational studies.

## National studies on the impact on firearm legislation of violence

Of the six included studies, four discuss the impact of legislations that restrict or limit the access to firearms on firearm-related violence on a national level [18, 20–22]. Two studies [20, 22] assess the impact of a singular law: a 1997 law adapted in Austria, that introduced stricter controlling mechanisms, such as background checks and mental health checks, the process of obtaining a firearm, as well as safe storage regulations. Both studies use official statistics as data sources and operationalize the measurements of availability of firearms and firearm-enabled violence using the same proxies: number of licenses for firearms and homicide by gunshot respectively. Consequently, the outcome of the studies regarding the impact of the law after 1997 on national firearm homicide rates are similar: both studies report a decline of firearm homicides post-regulation ranging from 9,6% [22] for the period of 1998 to 2008 to 9,9% [20] between 1998 and 2005. The change in rates is statistically significant in both of the studies. Additionally, König et al. examine the percentage of firearm enabled homicides in relation to the total number of homicides and found a decrease from 9% in 1998 to 2,6% in 2008 [22]. After 2008, the authors see the decline reversed into an increase of firearm homicides. They ascribe this rise to the economic crisis, which in turn led to an influx of migrants and an increase of issues of firearm licenses. Kapusta and colleagues also report a decline in firearm licenses after 1998 [20]. As none of the confounding variables used in both studies–unemployment rate, alcohol consumption and proportion of young men–show any significant effects, authors of these two studies cautiously provide support for the theory that higher availability of firearms is connected to increased violence.

**Table 2. Study design, level of analysis, operationalizations for gun violence and availability and outcome of included studies.**

| Study | Study Design | Level of Analysis | Operationalization (Gun) Violence | Operationalization Gun Availability | Outcome |
|---|---|---|---|---|---|
| Duquet & van Alstein (2015) [1] | Cross-sectional, correlational analysis | Cross-National, aggregated data | Gun related homicide according to WHO codes | Civilian firearm ownership rate | The higher the availability of firearms, the more women are killed by firearms; other relationships statistically insignificant |
| Gjertsen et al. (2014) [18] | Cross-sectional time series analysis | National, non-aggregated data | Accidental + homicide firearm related deaths according to WHO codes | Firearm restricting regulations | Removing firearms from private homes of National Guards associated with reduced firearm homicides |
| Hurka & Knill (2018) [19] | Cross-sectional time series analysis | Cross-National, aggregated data | Gun related homicide according to WHO codes | Firearm restricting regulations | Stricter firearm policies associated with less firearm- and non-firearm homicides |
| Kapusta et al. (2007) [20] | Cross-sectional time series analysis | National, non-aggregated data | Gun related homicides | Number of firearm licences | Stricter firearm policies is associated with a decrease of firearm licenses and gun homicides in Austria |
| Killias & Haas (2002) [21] | Cross-sectional multivariate regression analysis | National, non-aggregated data | Self-reported purposeful infliction of injury to another person | Number of handguns owned; Number of rifles owned; Frequency of carrying a weapon | Both owning and carrying a firearm increased the risk of injuring another person intentionally. |
| König et al. (2018) [22] | Cross-sectional times series analysis | National, non-aggregated data | Gun related homicides | Number of firearm licences | Stricter firearm policies is associated with a decrease of firearm licenses and gunhomicides in Austria |

Gjertsen et al. examine the impact of four different singular laws in Norway that were implemented between 1986 until 2003, aimed at regulating access to firearms by implementing new tests for hunters, permissions by police as a requirement for obtaining a shotgun, safe storage regulations and the removal of military firearms from private homes [18]. Because of a low incidence of female victims in relation to firearm deaths, the authors examined only male firearm deaths (N = 276) between 1969 and 2009. The only significant findings in relation to firearm homicides are found after the implementation of the latest firearm restricting regulation in 2003, that removed military firearms from private homes. After the implementation of the regulation, firearm homicides decreased by 64%. Rates of non-firearm homicides show no significant changes in the same period. The authors warn that their findings should not be overinterpreted, as the law under investigation targeted a very specific subgroup of Norwegian National Guards storing their weapons at home, and other factors not included in the analysis, such as changes in perceptions and behaviour regarding gun safety could have had an additional effect on fatal firearm violence in Norway.

## Cross-national study on the impact of firearm legislation on violence

Hurka and Knill assess a potential impact of firearm regulations regarding availability and firearm enabled violence [19]. They examine firearm homicide in 16 Western European countries over a timespan of thirty years. In contrast with the previously discussed studies, Hurka and Knill created an index to assess the level of restrictiveness in firearm regulations across the 16 countries included in the sample, thereby using an aggregated measure of firearm control policies, rather than assessments of individual regulations [19]. Through a cross-sectional time-series analysis, the authors conclude that the higher the level of restrictiveness regarding firearm control, the fewer firearm and non-firearm enabled homicides take place. They further specify that a difference of one value on their firearm control index (ranging from one to three) makes a difference of 0.2 homicides per 100.000 citizens a year. The control variables unemployment and urban population showed an increasing (0.01) and decreasing (-0.0) effect on firearm homicides respectively. Finally, they not only conclude that more restrictive firearm policies are associated with fewer homicides committed with firearms, but also that it is not likely that potential homicide offenders would switch to a different kind of weapon, given that stricter gun policies also appeared to have a decreasing effect on the overall homicide rate.

## (Cross-) national correlational study

The remaining studies that fit our eligibility criteria do not assess the impact of firearm regulations but use a proxy for firearm availability to assess its effect on firearm-enabled violence using correlational analyses. Duquet and van Alstein include data from 2007 for a broad sample of 33 European nations [1]. Data on firearm homicides are derived from the WHO's Detailed Mortality Database, whereas estimates for civilian firearm ownership from the Small Arms Survey serve as a proxy for firearm availability. The correlational analysis between civilian firearm ownership and firearm homicides reveals a moderate positive, yet statistically insignificant, relationship–both for firearm homicides and the overall homicide rate. Only when distinguishing between gender of firearm homicide victims, a significant moderate relation exists, suggesting that the greater the availability of firearms, the more women are killed by firearms within the countries under study. The authors suggest that this shows a specific effect of firearm availability on fatal domestic violence [1]. Other explanatory variables that could affect firearm homicides were not included in the analysis.

Killias and Haas, on the other hand, used confidential survey responses from a Swiss sample of male army recruits to assess whether owning a handgun or rifle, and carrying a weapon on a

regular basis had an impact on committing a violent act against another person [21]. The descriptive results indicate that participants in the survey who own a handgun have injured (10,7%) and shot at other people (4,4%) more often than non-owners of handguns (2,1%; 0%). Logistic regression analysis further supports the hypothesis that an increased number of handguns owned significantly increases the likelihood of inflicting injury on someone else intentionally (OR = 1,024). Analyses also reveal that every additional handgun owned raises the risk for violent incidents by 60 percent. Again, similar to previous studies presented in this review, the authors suggest that restricting access to firearms might lead to an overall decrease of violence committed with this type of weapon [21].

## Conclusion and discussion

This systematic review sought to assess all studies that statistically examine a potential link between the prevalence and availability of firearms with criminal forms of violence committed with firearms. A search of nine relevant databases revealed six studies that matched all eligibility criteria for this review. Four of these studies examined the impact of firearm-restricting regulations on the rate of firearm homicides, and two others used proxies for firearm availability to conduct correlational analyses with rates of firearm and non-firearm homicides.

Findings differ: whereas a firearm restricting regulation in Austria was associated with a decrease of almost 10 percent in firearm homicides in the following 10 years [20, 22], Norwegian scholars concluded that similar laws had little to no significant effect on a specific target group of male National Guards who stored their firearms at home [18]. Hurka and Knill's comparative analysis of Western European nations show that more restrictive regulations concurred with less firearm homicides, as well as a lower homicide rate overall [19]. In a correlational study, Duquet and van Alstein found no significant correlations between those two variables, except for female victims with fatal gunshot wounds [1]. Similar results were found by Killias et al. who sought to correlate homicide data with ownership rates of firearms from European and other nations [23]. Yet, the study by Killias and Haas reveal that owning a handgun significantly increases the risk of committing violent offences. Even more so, their data show that owning several handguns further elevates the risk by 60 percent per handgun owned [21].

One explanation for these varying results lies in the lack of comparability between the studies. International frameworks for firearm legislations, such as the firearm directives set out by the European Union [24], do exist. However, differences in implementations of those regulations on a national level, as well as cultural- and socio-economic backgrounds with regards to the use of firearms and crime in general lead to incongruent legislations across nations in Europe [25]. Another factor inhibiting the comparability of these studies is the range of operationalizations for availability of firearms. Even though various firearm directives by the EU encouraged all Member States to file and register civilian firearm ownership, European nations do not have identical systems to register legal firearm possession by civilians [26]. Moreover, in cases where reliable national registers exist, the prevalence of illegal firearms is not accounted for. Therefore, cross-national studies have diverted to using different proxies for firearm ownership and availability, such as survey data from the International Crime Victims Survey [23, 27, 28], firearm suicide rates [29, 30], accidental firearm death rates [30, 31], or the often-cited Cook's Index, which uses the average of the percentages of US suicides and homicides committed with firearms to estimate levels of ownership [32, 33]. In the two cross-national correlational studies included in this systematic review, three different proxies for firearm availability were used: accidental firearms deaths, suicides by gunshot and rate of civilian firearm ownership as reported by the Small Arms Survey, which in turn is based on multiple sources such as national registries, population surveys and expert estimates [34]. Although

these three proxies have been used by previous published studies, it should nonetheless be noted that the validity of many proxies for firearm prevalence has been questioned due to limitations of each proxy [for an overview, see 35]. As such measurement biases might have impacted the results, they need to be considered, especially when these studies are used by policymakers and practitioners to address the public health issue of firearm-related violence.

Overall, this systematic review highlights a lack of available studies based exclusively on European data. That is, however, not to say that the link between firearm prevalence and firearm violence has not been addressed in academic literature at all. There are number of widely cited cross-national correlational studies that include European data [23, 28, 29, 33, 36–38]. Yet, those studies also include data from other non-European nations, often the US, Canada, Japan or Australia, in their statistical correlational analysis, which makes it impossible to reveal findings based on European data only. Data from other included non-European countries might influence the overall results, considering cross-national large differences in firearm legislations and gun cultures [35]. Moreover, these studies show similar methodological weaknesses in terms of operationalizations and a lack of control for other factors influencing homicide rates [2, 7]. Based on this assessment, a systematic review including these cross-national studies with non-European data would encounter similar limitations regarding comparability of studies.

Nonetheless, a few results from relevant global studies should be discussed, in particular in relation to the findings of the six included studies. Similar to results presented in Duquet and van Alstein's study [1], Killias and colleagues found a strong positive correlation between firearm availability–operationalized as gun ownership according to the international victimization survey–and female gun homicide victimization, even when outliers are removed (Estonia, Malta, USA) [23]. In addition, both studies found no such significant correlation between firearm availability and male firearm homicide victimization [1, 23]. Other global studies have not differentiated between the gender of the victims of gun violence. Moreover, similar to the European findings discussed above, global studies show no conclusive, but varying results, ranging from no significant [23], to positive [28] as well as negative correlational [39] links between firearm availability and violent death by firearms. Although we have not conducted a systematic review of all global studies investigating the link between firearm availability and firearm violence, these varying results in the European context and beyond underline the importance of conducting in-depth, rigorous research that also include other factors relating to socio-economic and cultural factors that determine the context in which firearm violence takes place.

Next to international studies using non-European data, the findings of this systematic review can be complemented with other studies addressing the link between firearms and violence with a descriptive non-statistical approach. In doing so, several studies have linked an increase of (mostly illegal) firearms on the national level to an increase of firearm homicides or other types of crimes [40, 41]. In Sweden, for example, the percentage of firearm homicides in relation to other homicides has increased significantly, and a number of Swedish researchers have associated this increase with the rise of motorcycle- and other types of gangs who not only commit the crime, but also bring more weapons into the country [41–43]. In Switzerland, on the other hand, many firearm homicides take place in the private home in the context of domestic disputes. Killias and Markwalder relate this trend to a Swiss law that allowed (former) soldiers to keep their weapons after service at home [44]. Unfortunately, a small sample size did not allow the researchers to assess whether changes to the size of the army and therefore number of guns stored in private homes affected the homicide rate.

Even though such descriptive studies offer valuable insights into the context in which firearm violence takes place, they cannot sufficiently address the lack of insights into the association between firearm prevalence and–victimization. The main finding of our systematic

review regarding firearm availability and–interpersonal violence in Europe is that empirical studies are still rare and existing studies lack comparability due to both a national focus and to variations in measurements of firearm availability. Future empirical research should not only address this research gap but should also critically engage with the identified methodological difficulties, for example by evaluating various proxies and their reliability for measuring firearm availability based on European data. Moreover, future research should overcome some of the limitations of this systematic review. For example, we could only include studies that were published in English, German or Dutch. We cannot rule out that there are studies in other languages that would have been relevant for this review. Another limitation concerns the indirect publication bias present in this systematic review, as we only included published studies. To address this issue, we decided to include grey literature as well. Future research examining the link between firearm availability and violence should further pay specific attention to Eastern-European countries, which are now grossly underreported, as well as to types of violence other than homicide, such as non-fatal assaults, or robberies. More specifically, such research would benefit from addressing the methodological shortcomings of existing research by empirically testing the assumption that availability of firearms is associated with forms of firearm violence rather than relying on descriptive statistics. Furthermore, they can take socio-economic and cultural factors into account. In addition, an empirical examination of the validity of proxies used to measure firearm availability, following the example of Cook's index [32], could form the basis for above proposed empirical research and also increase comparability across studies. Only if those challenges and gaps are addressed will future studies become more comparable and valuable for public health and criminal justice researchers, policymakers and practitioners alike.

## Supporting information

**S1 File. PRISMA checklist.** Preferred reporting items for systematic reviews and meta-analyses (PRISMA) checklist.
(DOC)

**S2 File.**
(XLSX)

## Author Contributions

**Conceptualization:** Katharina Krüsselmann, Pauline Aarten, Marieke Liem.

**Data curation:** Katharina Krüsselmann.

**Formal analysis:** Katharina Krüsselmann.

**Funding acquisition:** Marieke Liem.

**Investigation:** Katharina Krüsselmann.

**Methodology:** Katharina Krüsselmann, Pauline Aarten, Marieke Liem.

**Project administration:** Katharina Krüsselmann, Pauline Aarten, Marieke Liem.

**Supervision:** Pauline Aarten, Marieke Liem.

**Validation:** Pauline Aarten, Marieke Liem.

**Writing – original draft:** Katharina Krüsselmann.

**Writing – review & editing:** Katharina Krüsselmann, Pauline Aarten, Marieke Liem.

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
