## [Decision Letter · Decision Letter 0]

4 Nov 2020

PONE-D-20-22981

Firearms and violence in Europe - A systematic review

PLOS ONE

Dear Dr. Krüsselmann,

Thank you for submitting your manuscript to PLOS ONE. After careful consideration, we feel that it has merit but does not fully meet PLOS ONE’s publication criteria as it currently stands. Therefore, we invite you to submit a revised version of the manuscript that addresses the points raised during the review process.

I have read the reviewer's comments and find them to be accurate, important, and constructive. I do want to bring your attention to Reviewer 2's encouragement to look at suicides and accidental shootings. I agree with the reviewer that it would be a very valuable thing to do. However, if it is outside the articulated scope of your study then PLOS will not require it since the reviewer's comment, as written, does not appear to be strictly related to PLOS publication criteria. Also, please read the PLOS Data Policy and make sure your submission comports with the requirements.

We look forward to receiving your revised manuscript.

Kind regards,

Chad M. Topaz

Academic Editor

PLOS ONE

2. Please include additional information about your study quality assessment method. In your manuscript, you refer to ref. 13 for more information, however, If materials, methods, and protocols are well established, authors may cite articles where those protocols are described in detail, but the submission should include sufficient information to be understood independent of these references (https://journals.plos.org/plosone/s/submission-guidelines#loc-materials-and-methods).

3.We note that you have indicated that data from this study are available upon request. PLOS only allows data to be available upon request if there are legal or ethical restrictions on sharing data publicly. For information on unacceptable data access restrictions, please see http://journals.plos.org/plosone/s/data-availability#loc-unacceptable-data-access-restrictions.

Reviewers' comments:

Reviewer's Responses to Questions

**Comments to the Author**

1. Is the manuscript technically sound, and do the data support the conclusions?

Reviewer #1: Yes

Reviewer #2: Yes

2. Has the statistical analysis been performed appropriately and rigorously? 

Reviewer #1: N/A

Reviewer #2: Yes

3. Have the authors made all data underlying the findings in their manuscript fully available?

Reviewer #1: Yes

Reviewer #2: No

4. Is the manuscript presented in an intelligible fashion and written in standard English?

Reviewer #1: Yes

Reviewer #2: Yes

5. Review Comments to the Author

Reviewer #1: The purpose of this paper was to both summarize the European studies on how firearm availability connects to rates of interpersonal violence and to inform the potential effects of firearm restrictive legislations and future trends. It was an informative and thorough search of the literature that exposed the need for further work in this field. The writing is clear, concise, and explains the literature in appropriate detail. The first goal is met in that only six studies were appropriate for inclusion in this study. The second goal is not as strongly supported due to the small sample size of this study. Therefore, I recommend major revisions.

The major issues are as follows:

1. There is a need to either restate the second goal of the paper (lines 68-71), or to add more information that supports it. Because the papers have few common findings, it does not seem reasonable to use this work to inform potential effects and future trends. The reasons behind this incongruence are explained in the discussion. If this is still a goal of this manuscript, then far more information needs to be added and synthesized to make specific statements. Conversely, if this goal can be restated, this is no longer a major issue.

2. Another major issue is the number of studies reviewed. A sample size of six makes it difficult to accept any conclusions, even if the conclusions are that there are not enough studies. This can be remedied by either relaxing the inclusion criteria and including more studies or discussing some of the results from excluded studies to give more support to specific conclusions.

In terms of minor revisions:

1. For the Introduction (lines 47-71), include more information regarding firearm use in Europe. For non-European readers, it would be helpful to understand more about the legislation that is in place and the culture surrounding firearms in Europe. Some of this is discussed in lines 258-262, but may be better suited as background information. Similarly, include more information about past analyses of legislation. Particularly, The Science of Gun Policy (Smart 2020) conducts these kinds of studies in the United States and there are likely others. Highlighting some of these past works and the methods used to synthesize multiple papers or pieces of legislation will give support to your claims.

2. For Table 1, add the title of the manuscript to the table. Even though the bibliography contains this information, readers are more likely to look at the studies if the title is readily available.

3. Clarify the differing levels of analysis and potentially define the study design types (Table 1, Table 2, lines 169-173). Especially when discussing levels of analysis, some readers may understand ``individual'' to mean specific people and it is unclear whether this is the case in your analysis.

4. Within the conclusion and discussion, present some of the ways in which future studies could meet inclusion criteria for a future review. There are many mentions of the need for consistency across studies, but tying back in the reasons why certain studies were or were not included in your analysis would give more guidance to researchers in the field.

Thank you for your work on this manuscript and your contribution to the field!

Reviewer #2: Please see attached. I'm copying and pasting the first bit to get over 200 characters. This paper fills an important gap in the literature, by conducting a systematic review, following the PRISMA guidelines, of the corpus of papers that use data to study the Firearms Availability Hypothesis in Europe. This hypothesis states that greater availability of firearms is associated with more incidents of firearm-enabled criminal violence. The authors exhaustively searched nine databases and some 8,000 papers to find all papers satisfying their desiderata. The result is six papers that address this question, using data, during the appropriate time frame, using control variables, and producing robust results. The authors do an excellent job explaining their methodology, describing these six papers, distilling conclusions, and making recommendations for future work. I expect that future papers will build on and cite this paper.

I have two substantive questions/comments for the authors, and then a small number of line by line comments.

(1) The authors mention that they ruled out 25 papers where they were unable to get a full-text copy. Have they succeeded in getting any of these in the months since? It seems like Feb - April was a hard time everywhere, and so the authors they emailed might have dropped the ball on writing back. Were those authors ever emailed again?

(2) In the Outcomes column of Table 2, many of the papers are summarized using overly causal language. For example, when describing reference [14], the table writes ``Removing firearms from private homes of National Guards reduces firearm homicides'' but [14] uses less causal language, saying the laws ``could have contributed.'' Similarly, in describing [15] the table says ``Stricter firearm policies lead to less firearm and non-firearm homicides'' but the methods of [15] (linear and Poisson regression) do not support causal conclusions. Similarly, ``lead to'' is used in the description of [16] and [18], while for the description of [17], ``Both owning and carrying a firearm increased the risk of injuring another person intentionally'' again suggests a causal link. In all these cases, the authors of those studies are careful to avoid overly causal language. Furthermore, all six of the study designs (except possibly the quasi-experimental design of [17]) do not seem capable of proving causation. I encourage the authors to modify their language to strip out unsupported causal claims. For example, `lead to" can be replaced (in both Table 2 and the subsequent discussion of the six papers) with ``is associated with."

As the authors themselves write in the first paragraph of the introduction, ``causal links between the prevalence of firearms and violence remain unclear." It's best not to accidentally insert causal claims where they were not supported by the papers in question. For example, [14] is an observational study. Most of the six papers use time-series methods and observe that gun violence dropped when new laws were put into effect. But this approach ignores the possibility that culture is a confounding variable that causes both the change in laws and the drop in gun violence.

My last comment before moving on to the line-by-line remarks is that I want to encourage the authors to take up a similar literature review for the interplay between firearm availability and suicide rates, as well as accidental shootings. Since you've already done the first step, of combing through all the papers, this would be easier for you than for other researchers and it's arguably just as important as the systematic review you've just completed.

There are also attached line-by-line remarks.

6. PLOS authors have the option to publish the peer review history of their article (what does this mean?). If published, this will include your full peer review and any attached files.

Reviewer #1: **Yes: **Shelby M Scott

Reviewer #2: No

---

## [Author Response · Author response to Decision Letter 0]

22 Jan 2021

Reviewer 1

Comment 1: There is a need to either restate the second goal of the paper (lines 68-71), or 

 to add more information that supports it. Because the papers have few 

 common findings, it does not seem reasonable to use this work to inform 

 potential effects and future trends. The reasons behind this incongruence are 

 explained in the discussion. If this is still a goal of this manuscript, then far 

 more information needs to be added and synthesized to make specific 

 statements. Conversely, if this goal can be restated, this is no longer a major 

 issue.

Response: First of all, thank you for your clear comments throughout your review. It is 

 indeed clear that we did not meet our second goal, given the low quantity of 

 eligible studies and their methodological characteristics, which did not allow 

 for us to make any clear conclusions regarding the effects of firearm legislation. 

 We have restated our goal to match the information gained from this systematic 

 review more clearly.

Changes: We have changed our second goal to: ‘inform researchers, practitioners and 

 policymakers in the domains of public health and criminal justice about the 

 current state of knowledge regarding the association between firearm 

 availability and violence, remaining lacunae of knowledge to be filled by and 

 methodological challenges to be addressed by future research’. 

The text has been adapted in the introduction (lines 79-83) and the conclusion.

Comment 2: Another major issue is the number of studies reviewed. A sample size of six 

 makes it difficult to accept any conclusions, even if the conclusions are that 

 there are not enough studies. This can be remedied by either relaxing the 

 inclusion criteria and including more studies or discussing some of the results 

 from excluded studies to give more support to specific conclusions.

Response: We agree that the low N of eligible studies presents a disadvantage, as already 

 stated in the response to your first comment. In the preparation of the search 

 protocol, we have tried different inclusion criteria and agreed on the necessary 

 practical, as well as logical inclusion criteria, such as location and timeframe 

 that allow us to answer the research question. To overcome some of the 

 limitations of the low N, we placed our findings in a broader context by 

 comparing them with findings from international studies that include, amongst 

 others, European data in the discussion. To emphasize the relevance of those 

 studies, we now added a more concrete summary of those international studies 

 in the discussion, as well as a short comparison of the findings from the global 

 studies with the included European studies. A challenge in such a comparison 

 – yet at the same time important conclusion – is that comparisons are difficult 

 due to the varying results found in each of the studies, mainly due to socio-

 economic and cultural differences between the countries under study. We have 

 now explicitly stated this in the discussion. Moreover, we hope that the low 

 number of empirical studies will show the need for methodologically robust 

 research on this topic. 

Changes: We have added a paragraph (lines 315-329) in the discussion with some 

 additional observations. 

Comment 3: For the Introduction (lines 47-71), include more information regarding firearm 

 use in Europe. For non-European readers, it would be helpful to understand 

 more about the legislation that is in place and the culture surrounding firearms 

 in Europe. Some of this is discussed in lines 258-262, but may be better suited 

 as background information. Similarly, include more information about past 

 analyses of legislation. Particularly, The Science of Gun Policy (Smart 2020) 

 conducts these kinds of studies in the United States and there are likely others. 

 Highlighting some of these past works and the methods used to synthesize 

 multiple papers or pieces of legislation will give support to your claims.

Response: Thank you for this valuable comment. 

Changes: We have added a summary (lines 59-74) of relevant research on gun 

 culture and -legislation in Europe to the introduction. 

Comment 4: For Table 1, add the title of the manuscript to the table. Even though the 

 bibliography contains this information, readers are more likely to look at the 

 studies if the title is readily available.

Response: We agree. 

Changes: Titles of studies are added in table 1.

Comment 5: Clarify the differing levels of analysis and potentially define the study design 

 types (Table 1, Table 2, lines 169-173). Especially when discussing levels of 

 analysis, some readers may understand ``individual'' to mean specific people 

 and it is unclear whether this is the case in your analysis.

Response: This is an important comment, thank you. To clarify the kind of data used in the 

 different studies, and to avoid misunderstanding, we have adapted the 

 information presented in table 2 to reflect whether a study was conducted 

 nationally vs. cross-nationally, using aggregated vs. non-aggregated data. 

 Moreover, we have added a column in table 2, describing the research design 

 of each study. 

Changes: In table 2, the level of analysis has been adapted. A clarification has also been 

 added in lines 170-178. A column ‘study’ design was added in table 2.

Comment 6: Within the conclusion and discussion, present some of the ways in which 

 future studies could meet inclusion criteria for a future review. There are many 

 mentions of the need for consistency across studies, but tying back in the 

 reasons why certain studies were or were not included in your analysis would 

 give more guidance to researchers in the field.

Response: In addition to the suggestions for future research mentioned in the 

 original manuscript, we have added two additional suggestions, which – we 

 believe – would increase the quality of future research: first, although various 

 proxies of firearm availabilities are used, there is no examination of the validity 

 of these proxies. Therefore, we suggest that future studies should focus on 

 available proxies, such as data from the Small Arms Survey, the 

 International Crime Victim Survey or European firearm suicide data. Such a 

 research has been carried out in the past in the US context. Our second 

 suggestion is that more studies need to carry out empirical tests to assess the 

 link between firearm availability and -violence. Although various studies on 

 fatal and non-fatal violence in Europe mention firearm availability as a factor, 

 most assumed the association between availability and violence, rather than 

 testing such a link.

Changes: We have added above described suggestions for future research in lines 357-

 365 in the revised manuscript.

Reviewer 2

Comment 1: The authors mention that they ruled out 25 papers where they were unable to 

 get a full-text copy. Have they succeeded in getting any of these in the months 

 since? It seems like Feb - April was a hard time everywhere, and so the 

 authors they emailed might have dropped the ball on writing back. Were those 

 authors ever emailed again? 

Response: This is a very good point. The pandemic has changed all of our private and 

 professional rhythms and it would be more than understandable if our 

 contacted authors were not able to respond to our initial email. Therefore, we 

 have send out another email again, on November 10th, as well as follow-up 

 reminder on November 24th. As a result, we have received two more studies, 

 which we have assessed according to our eligibility criteria. Unfortunately, we 

 had to exclude both studies on the merit that they did not (exclusively) include 

 European data. We have adapted the text, as well as the PRISMA flow chart 

 accordingly. 

Changes: Numbers of studies received and excluded were adapted in the text (lines 142-

 147) and the PRISMA flow chart.

Comment 2: In the Outcomes column of Table 2, many of the papers are summarized using 

 overly causal language. For example, when describing reference [14], the table 

 writes “Removing firearms from private homes of National Guards reduces 

 firearm homicides” but [14] uses less causal language, saying the laws “could 

 have contributed.” Similarly, in describing [15] the table says “Stricter firearm 

 policies lead to less firearm and non-firearm homicides” but the methods of 

 [15] (linear and Poisson regression) do not support causal conclusions. 

 Similarly, “lead to” is used in the description of [16] and [18], while for the 

 description of [17], “Both owning and carrying a firearm increased the risk of 

 injuring another person intentionally” again suggests a causal link. In all these 

 cases, the authors of those studies are careful to avoid overly causal language. 

 Furthermore, all six of the study designs (except possibly the quasi-

 experimental design of [17]) do not seem capable of proving causation. I 

 encourage the authors to modify their language to strip out unsupported causal 

 claims. For example, ‘lead to” can be replaced (in both Table 2 and the 

 subsequent discussion of the six papers) with “is associated with.” As the 

 authors themselves write in the first paragraph of the introduction, “causal 

 links between the prevalence of firearms and violence remain unclear.” It’s 

 best not to accidentally insert causal claims where they were not supported by 

 the papers in question. For example, [14] is an observational study. Most of 

 the six papers use time-series methods and observe that gun violence dropped 

 when new laws were put into effect. But this approach ignores the possibility 

 that culture is a confounding variable that causes both the change in laws and 

 the drop in gun violence. 

Response: This is a very valuable comment, thank you. Although we have tried to adapt 

 the language used by the authors of the included studies, we see that some of 

 the statements made in our review require careful revision to avoid making 

 causal claims that are not supported by the data. We have critically and 

 carefully reviewed the language used throughout and adapted where necessary.

Changes: Language has been adapted throughout the analysis, to avoid overly causal 

 claims that are not supported. For example in table 2. 

Comment 3: My last comment before moving on to the line-by-line remarks is that I want 

 to encourage the authors to take up a similar literature review for the interplay 

 between firearm availability and suicide rates, as well as accidental shootings. 

 Since you’ve already done the first step, of combing through all the papers, 

 this would be easier for you than for other researchers and it’s arguably just as 

 important as the systematic review you’ve just completed. 

Response: We agree that such a review would be a great addition to the body of scholarly 

 work. For our study, however, we have decided to exclude accidents and 

 suicides, due to our expectation that those forms of violence have different 

 underlying causes. However, we will keep the extracted literature and hope to 

 continue and explore the association of firearms and suicides and accidents in 

 the future .

Changes: No changes made.

Comment 4: Around line 116, the authors write that their approach was inspired by [2] and 

 [6]. It would be good to say a word here about why this paper is different from 

 [2] and [6], and still needed despite [2] and [6] already doing a systematic 

 review of literature related to firearm availability. I’m guessing it’s because 

 [2] includes data from outside Europe and does not include control variables, 

 and [6] includes data from outside Europe, but this would be good to say. 

Response: A good point. We have added the reason to our text. 

Changes: See line 129-130. 

Comment 5: 

• Line 162 has a dash that it doesn’t seem to need. 

• Same in the Outcomes column for the Kapusta paper and for the K ¨onig paper. 

• Same on lines 304 and 305. 

• Line 241: “This systematic review sought to assessing all studies” 

• Line 262: should “incongruent legislations” be “legislation” instead? I’m not sure. 

• Line 282, should “in academic literature” be “in the academic literature”? 

• Line 295, I think either the dash after ‘motorcycle’ should be dropped, or you have to add another dash after ‘gangs’ on line 296. 

Response: Thank you for your detailed observations. The dashes were included to indicate 

 (for example in line 304) that we want to make a statement about firearm 

 homicides, not homicides in general. We left ‘legislations’ instead of 

 ‘legislation’, given that many European countries – namely EU member states 

 – follow the same framework of regulation, yet national differences exist.

Changes: Adapted in the text.

---

## [Decision Letter · Decision Letter 1]

5 Mar 2021

PONE-D-20-22981R1

Firearms and violence in Europe - A systematic review

PLOS ONE

Dear Dr. Krüsselmann,

Thank you for submitting your manuscript to PLOS ONE. After careful consideration, we feel that it has merit but does not fully meet PLOS ONE’s publication criteria as it currently stands. Therefore, we invite you to submit a revised version of the manuscript that addresses the points raised during the review process.

The revised version of the paper has been well-valued by our reviewers. Thanks for the clarity and precision of your amendments. However, one oft referees would like to ask for some further minor changes for accepting the paper. Please resubmit the revised version of the manuscript with these modifications; if the quality of the amendments and the set of responses given by you are OK, I will proceed to accept the paper without requiring a new round of reviews.

We look forward to receiving your revised manuscript.

Kind regards,

Sergio A. Useche, Ph.D.

Academic Editor

PLOS ONE

Journal Requirements:

Reviewers' comments:

Reviewer's Responses to Questions

**Comments to the Author**

1. If the authors have adequately addressed your comments raised in a previous round of review and you feel that this manuscript is now acceptable for publication, you may indicate that here to bypass the “Comments to the Author” section, enter your conflict of interest statement in the “Confidential to Editor” section, and submit your "Accept" recommendation.

Reviewer #1: (No Response)

Reviewer #2: All comments have been addressed

2. Is the manuscript technically sound, and do the data support the conclusions?

Reviewer #1: Yes

Reviewer #2: Yes

3. Has the statistical analysis been performed appropriately and rigorously? 

Reviewer #1: Yes

Reviewer #2: Yes

4. Have the authors made all data underlying the findings in their manuscript fully available?

Reviewer #1: Yes

Reviewer #2: Yes

5. Is the manuscript presented in an intelligible fashion and written in standard English?

Reviewer #1: Yes

Reviewer #2: Yes

6. Review Comments to the Author

Reviewer #1: The manuscript entitled, ``Firearms and Violence in Europe'' has been improved greatly by the authors and I appreciate the attention given to initial calls for revision. The purpose of this article is to summarize the existing studies of firearm violence in Europe, but has been revised to highlight the need for further studies before specific conclusions can be drawn. With the revisions that have been made, I recommend acceptance with minor revisions. The minor revisions are listed below.

Minor Revisions

1. Line 49: Put this statistic in per capita form and potentially compare it to the continental United States or another similar region. As a researcher from the U.S., 7,000 seems like a small number, but is even smaller (relatively) when considering the population of continental Europe. Comparing it to another country will put it into perspective for readers.

2. Line 60: Change ``vultures'' to ``cultures.'' This is an understandable typo, but I wanted to be sure it was highlighted!

3. Lines 157 - 160: italicize or bold good, fair, and poor when initially defining them. This will make it easier to find the definitions for readers as they work through the review.

4. General: Extend the captions for figures and tables. Add a brief statement about what should be taken away from each table and figure so that they would be able to stand alone, without the text of the main manuscript

Thank you for your willingness to address the concerns from the first reviews and for completing this necessary work that will add to the field of firearms violence research.

Reviewer #2: This paper is ready to be accepted. I did, however, spot a few more typos:

Line 61: please capitalize Constitution

Line 62: "baring" should be "bearing"

Lines 154-157: Did you mean to have (1)-(5) display as one long sentence, or did you mean to have one item per line? I think it would be easier to read if it was one item per line, but it's up to you.

Line 316: perhaps this colon should be a period since the next word is capitalized

Line 323: there is a hyphen before a comma that does not seem to belong

7. PLOS authors have the option to publish the peer review history of their article (what does this mean?). If published, this will include your full peer review and any attached files.

Reviewer #1: No

Reviewer #2: No

---

## [Author Response · Author response to Decision Letter 1]

5 Mar 2021

We have adapted all the recommended changes in the revised manuscript.

---

## [Editor Report · Decision Letter 2]

9 Mar 2021

Firearms and violence in Europe - A systematic review

PONE-D-20-22981R2

Dear Dr. Krüsselmann,

We’re pleased to inform you that your manuscript has been judged scientifically suitable for publication and will be formally accepted for publication once it meets all outstanding technical requirements.

Kind regards,

Sergio A. Useche, Ph.D.

Academic Editor

PLOS ONE

Additional Editor Comments (optional):

Dear Authors: thanks for the adequacy of the amendments made. I will now proceed to accept the paper in its present form.

---

## [Editor Report · Acceptance letter]

18 Mar 2021

PONE-D-20-22981R2 

Firearms and violence in Europe – A systematic review 

Dear Dr. Krüsselmann:

I'm pleased to inform you that your manuscript has been deemed suitable for publication in PLOS ONE. Congratulations! Your manuscript is now with our production department. 

Kind regards, 

on behalf of

Dr. Sergio A. Useche 

Academic Editor

PLOS ONE